# Comparison of 3 optimized delivery strategies for completion of isoniazid-rifapentine (3HP) for tuberculosis prevention among people living with HIV in Uganda: A single-center randomized trial

**Fred C. Semitala**[1,2,3‡], **Jillian L. Kadota**[4,5‡], **Allan Musinguzi**[2], **Fred Welishe**[2], **Anne Nakitende**[2], **Lydia Akello**[2], **Lynn Kunihira Tinka**[2], **Jane Nakimuli**[2], **Joan Ritar Kasidi**[2], **Opira Bishop**[2], **Suzan Nakasendwa**[6], **Yeonsoo Baik**[7], **Devika Patel**[8], **Amanda Sammann**[8], **Payam Nahid**[4,5], **Robert Belknap**[9], **Moses R. Kamya**[1,2], **Margaret A. Handley**[10,11], **Patrick PJ Phillips**[4,5], **Anne Katahoire**[12], **Christopher A. Berger**[4,5], **Noah Kiwanuka**[6], **Achilles Katamba**[13,14], **David W. Dowdy**[14,15‡], **Adithya Cattamanchi**[4,14,16‡*]

1 Makerere University, Department of Medicine, College of Health Sciences, Kampala, Uganda, 2 Infectious Diseases Research Collaboration, Kampala, Uganda, 3 Makerere University Joint AIDS Program, Kampala Uganda, 4 Center for Tuberculosis, University of California San Francisco, San Francisco, California, United States of America, 5 Division of Pulmonary and Critical Care Medicine, San Francisco General Hospital, University of California San Francisco, San Francisco, California, United States of America, 6 Department of Epidemiology and Biostatistics, School of Public Health, Makerere University College of Health Sciences, Kampala, Uganda, 7 Department of Biostatistics, Epidemiology, and Informatics, Perelman School of Medicine, University of Pennsylvania, Philadelphia, Pennsylvania, United States of America, 8 The Better Lab and Department of Surgery, San Francisco General Hospital, University of California San Francisco, San Francisco, California, United States of America, 9 Denver Health and Hospital Authority and Division of Infectious Diseases, Department of Medicine, University of Colorado, Denver, Colorado, United States of America, 10 Center for Vulnerable Populations, San Francisco General Hospital, University of California San Francisco, San Francisco, California, United States of America, 11 Department of Epidemiology and Biostatistics, University of California San Francisco, San Francisco, California, United States of America, 12 Child Health and Development Center, School of Medicine, Makerere University College of Health Sciences, Kampala, Uganda, 13 Clinical Epidemiology & Biostatistics Unit, Department of Medicine, Makerere University College of Health Sciences, Kampala, Uganda, 14 Uganda Tuberculosis Implementation Research Consortium, Walimu, Kampala, Uganda, 15 Department of Epidemiology, Johns Hopkins Bloomberg School of Public Health, Baltimore, Maryland, United States of America, 16 Division of Pulmonary Diseases and Critical Care Medicine, University of California Irvine, Irvine, California, United States of America

‡ FCS and JLK share first authorship on this work. DWD and AC are joint senior authors on this work.
* Adithya.cattamanchi@uci.edu

## Abstract

### Background

Expanding access to shorter regimens for tuberculosis (TB) prevention, such as once-weekly isoniazid and rifapentine taken for 3 months (3HP), is critical for reducing global TB burden among people living with HIV (PLHIV). Our coprimary hypotheses were that high levels of acceptance and completion of 3HP could be achieved with delivery strategies optimized to overcome well-contextualized barriers and that 3HP acceptance and completion

**Data Availability Statement:** All relevant data are within the manuscript and its Supporting Information files.

**Funding:** This study was supported by a grant from the US National Heart, Lung and Blood Institute (https://www.nhlbi.nih.gov/): NIH/NHLBI R01HL144406 (AC). The funder had no role in the study design, data collection and analysis, preparation of the manuscript or decision to publish.

**Competing interests:** A.S. and D.P. are human-centered design consultants for The Empathy Studio, LLC. D.P. is a human-centered design consultant for the Diversity Innovation Hub at Mt. Sinai. The other authors have declared that no competing interests exist.

**Abbreviations:** ALT, alanine aminotransferase; ART, antiretroviral therapy; AST, aspartate aminotransferase; CI, confidence interval; DOT, directly observed therapy; HCD, human-centered design; IQR, interquartile range; IVR, interactive voice response; PLHIV, people living with HIV; PR, prevalence ratio; RD, risk difference; SAT, self-administered therapy; TB, tuberculosis; TPT, TB preventive treatment; WHO, World Health Organization; 3HP, once-weekly isoniazid and rifapentine taken for 3 months.

would be highest when PLHIV were provided an informed choice between delivery strategies.

## Methods and findings

In a pragmatic, single-center, 3-arm, parallel-group randomized trial, PLHIV receiving care at a large urban HIV clinic in Kampala, Uganda, were randomly assigned (1:1:1) to receive 3HP by facilitated directly observed therapy (DOT), facilitated self-administered therapy (SAT), or informed choice between facilitated DOT and facilitated SAT using a shared decision-making aid. We assessed the primary outcome of acceptance and completion ($\geq$11 of 12 doses of 3HP) within 16 weeks of treatment initiation using proportions with exact binomial confidence intervals (CIs). We compared proportions between arms using Fisher's exact test (two-sided $\alpha = 0.025$). Trial investigators were blinded to primary and secondary outcomes by study arm. Between July 13, 2020, and July 8, 2022, 1,656 PLHIV underwent randomization, with equal numbers allocated to each study arm. One participant was erroneously enrolled a second time and was excluded in the primary intention-to-treat analysis. Among the remaining 1,655 participants, the proportion who accepted and completed 3HP exceeded the prespecified 80% target in the DOT (0.94; 97.5% CI [0.91, 0.96] $p < 0.001$), SAT (0.92; 97.5% CI [0.89, 0.94] $p < 0.001$), and Choice (0.93; 97.5% CI [0.91, 0.96] $p < 0.001$) arms. There was no difference in acceptance and completion between any 2 arms overall or in prespecified subgroup analyses based on sex, age, time on antiretroviral therapy, and history of prior treatment for TB or TB infection. Only 14 (0.8%) participants experienced an adverse event prompting discontinuation of 3HP. The main limitation of the study is that it was conducted in a single center. Multicenter studies are now needed to confirm the feasibility and generalizability of the facilitated 3HP delivery strategies in other settings.

## Conclusions

Short-course TB preventive treatment was widely accepted by PLHIV in Uganda, and very high levels of treatment completion were achieved in a programmatic setting with delivery strategies tailored to address known barriers.

## Trial Registration

ClinicalTrials.gov NCT03934931.

---

### Author summary

#### Why was this study done?

- There is limited evidence on strategies that achieve high acceptance and completion of tuberculosis (TB) preventive treatment in the context of routine HIV/AIDS care, despite recommendations from the World Health Organization for its scale-up in high-burden HIV/TB settings.

- Two previous implementation trials have assessed delivery of 12 weeks of isoniazid and rifapentine (3HP) by self-administered therapy (SAT) in high-burden settings and

found variable completion rates. In both studies, 3HP dosing and treatment supervision were conducted by research staff rather than by routine health workers.

### What did the researchers do and find?

- In a pragmatic trial,1,655 people living with HIV (PLHIV) at a high-volume clinic in Kampala, Uganda, received 3HP either by facilitated directly observed therapy (DOT) involving a pharmacy technician, facilitated SAT, or informed choice between facilitated DOT and facilitated SAT using a shared decision-making aid.

- We found high levels of 3HP completion in the context of routine HIV/AIDS care.

- The high treatment completion rates were independent of 3HP delivery strategy.

### What do these findings mean?

- High levels of 3HP treatment completion are achievable in routine programmatic settings when delivery strategies are optimized to overcome known barriers.

- A limitation of the study is that it was conducted at a single center. Further studies are needed to confirm the findings and the feasibility of the facilitation strategies in other settings.

## Introduction

Tuberculosis (TB) is a curable and preventable disease but remains a leading cause of death worldwide, especially among people living with HIV (PLHIV) [1]. TB preventive treatment (TPT) can reduce TB incidence by 30% to 50% [2] and the risk of death or severe illness by 35% among PLHIV [3]. Scaling up TPT is crucial to the END TB Strategy and global coverage has improved, particularly among PLHIV [4,5]. However, a recent study estimated completion of the conventional 6 to 9 months of daily isoniazid TPT regimen among PLHIV in 16 high-burden countries to be 66% overall and under 50% in 4 of the countries [6].

Short-course regimens for TB prevention, such as 3HP (once-weekly isoniazid and rifapentine for 12 weeks), are now available. The World Health Organization (WHO) recommends 3HP as an alternative to daily isoniazid, citing its shortened treatment duration, equivalent efficacy for TB prevention, and better tolerability and safety [7,8]. However, evidence remains limited on how to effectively deliver 3HP in a manner that achieves high acceptance and completion in real-world settings. In 2 previous implementation trials [9,10], trained research staff provided 3HP, rather than integrating it into routine care, and one found low (38%) completion rates with self-administered therapy at the only sub-Saharan African site included [9].

To assist National TB Programs in planning 3HP scale-up, we conducted the 3HP Options Trial. This pragmatic, 3-arm, type 3 hybrid effectiveness-implementation randomized trial [11] aimed to evaluate 3 facilitated strategies for delivering 3HP to PLHIV within routine HIV/AIDS care [12]. Our coprimary hypotheses were that, in a high HIV/TB burden setting, the proportion of PLHIV accepting and completing 3HP could exceed 80%, and this proportion would be highest among PLHIV randomized to the informed choice arm.

## Methods

### Ethics statement

The study protocol [12] was approved by ethical committees at the University of California San Francisco and Makerere University. Written informed consent was obtained from all study participants. An independent trial steering committee periodically oversaw trial conduct and approved protocol changes (registered at clinicaltrials.gov, NCT03934931).

### Study design and participants

We conducted this pragmatic, single-center, 3-arm, parallel-group randomized trial at the Mulago Immune Suppression Syndrome (HIV/AIDS) clinic in Kampala, Uganda. Eligible participants included PLHIV aged 18 years or older, receiving HIV/AIDS care at the clinic, and candidates for 3HP-based TB preventive treatment (S1 Text). We excluded people weighing <40 kilograms (as 3HP was not weight adjusted), initiating antiretroviral therapy (ART) within the previous 3 months, with documented clinical liver disease or history of alcohol abuse, with baseline serum alanine aminotransferase (ALT) or aspartate aminotransferase (AST) level exceeding 3 times the upper limit of normal, not intending to stay within 25 kilometers of the study clinic (to enable follow-up), without access to a mobile telephone or not willing to receive phone-based reminders (which would interfere with the facilitated self-administered therapy delivery strategy), or living with another household member already enrolled in the study.

Potential participants were recruited from the clinic waiting area through peer health educators or clinic providers. Interested PLHIV were referred to research staff for eligibility confirmation and written informed consent. The study design and results are reported following CONSORT 2010 guidelines [13].

### Randomization and masking

Participants were randomly assigned (1:1:1) to receive 3HP through facilitated directly observed therapy (DOT), facilitated self-administered therapy (SAT), or informed choice between facilitated DOT and facilitated SAT (Fig 1). Random permuted blocks of variable size (9, 12, and 15) with equal allocation were generated by the trial statistician. Sealed opaque envelopes containing individual random assignments were given to study nurses in multiple batches to minimize predictability. After informed consent, each participant selected and opened the top-most envelope to reveal their randomization assignment. Due to the intervention nature, participants and study staff were not masked to arm assignment. However, minimal interaction occurred between study staff and participants after enrollment; routine clinic staff provided 3HP doses, screened for adverse effects and made decisions about holding or discontinuing 3HP treatment. Throughout the trial, the principal investigators remained masked to randomization assignments.

### Interventions

Methods by which the delivery strategies were conceptualized, including results from formative research conducted with key local and national stakeholders, are detailed elsewhere [12,14–16]. Briefly, we used theory-informed frameworks, human-centered design (HCD) methods, and person-centered care principles to optimize 3HP delivery by SAT and DOT to address key capability, opportunity, and motivation barriers [17].

Facilitation strategies in all arms included standardized counseling and transport reimbursement for clinic visits. The value of transport reimbursement varied ($5 to $10 USD)

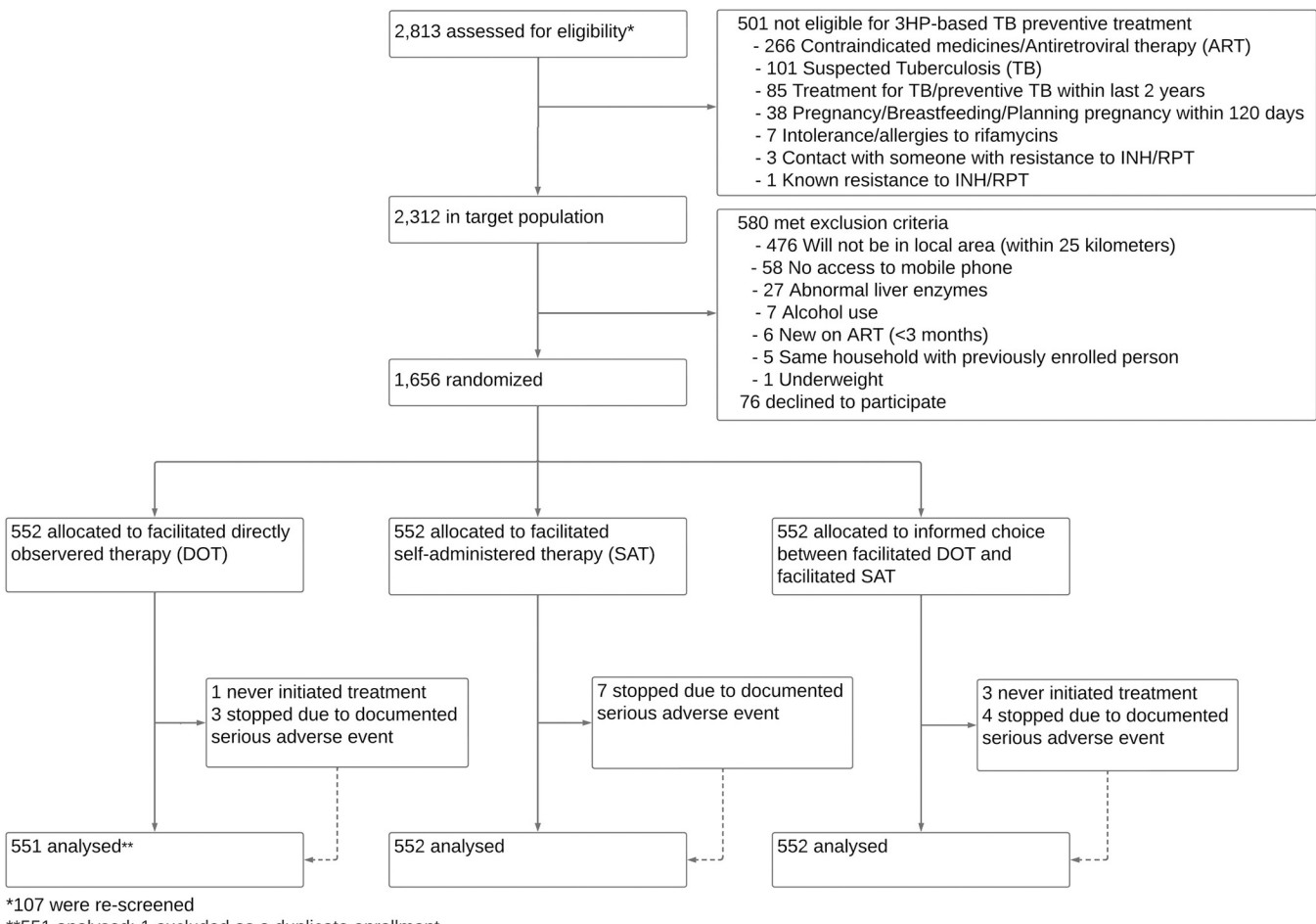

**Fig 1. Trial profile.** INH/RPT, isoniazid/rifapentine; 3HP, once-weekly isoniazid and rifapentine for 12 weeks.

based on COVID-19 impacts on public transportation costs during the study period. 3HP was provided using loose pills (six 150 mg rifapentine tablets, three 300 mg isoniazid tablets, and two 25 mg pyridoxine tablets) for the first 4 months of the trial and using fixed-dose combination pills (3 rifapentine 300 mg/isoniazid 300 mg tablets and two 25 mg pyridoxine tablets) for the remainder of the trial. Unique components of each delivery strategy are described below.

**Facilitated DOT.** Participants returned to the clinic pharmacy window weekly on their preferred day and time for a streamlined pharmacy visit, without the need to wait in the general queue. During each visit, a clinic pharmacy technician screened them for adverse events and TB symptoms (referring those who screened positive to a clinician), observed 3HP dose ingestion, and recorded dosing. The day prior to each weekly appointment, participants received a short, one-way automated interactive voice response (IVR) phone call (at no cost to participants) that included a motivational message that changed each week and a reminder to return to the clinic for their appointment. If the clinic appointment was missed, the one-way IVR message was repeated for up to 3 consecutive days.

**Facilitated SAT.** Participants received a pill pack codesigned with PLHIV using an HCD process [12]. The pill pack included individual plastic bags containing each weekly dose that contained (1) the correct number of 3HP pills for that week's dose and (2) a card insert

displaying a motivational message and a unique toll-free phone number participants were instructed to call after taking their weekly dose to confirm dosing. Once the number was called, the 99DOTS (Everwell Health Solutions, Bengaluru, India) platform automatically reflected the dose as taken and an individualized adherence calendar could be remotely accessed and monitored by clinic pharmacy technicians for 3HP adherence (**S1 and S2 Figs**). Participants were asked to return to the clinic for the sixth and 12th doses for pill counts, in-person dosing, and assessment of side effects. The day prior to their expected next dose, participants received a two-way automated IVR phone call that included a motivational message, a reminder to take their weekly dose, and the question "Are you well?", which enabled participants to respond affirmatively or negatively using the phone keypad. The IVR message was repeated for up to 3 consecutive days if there was no confirmation of dosing using the toll-free number. Pharmacy technicians were asked to call participants who reported feeling unwell to the two-way IVR message to conduct further assessments.

**Informed choice.** A research nurse used a counselling flipbook to provide a brief descriptive overview of facilitated DOT and facilitated SAT. Participants were then asked to state their preferred option considering their lifestyle and values with respect to 7 key concepts related to 3HP delivery (time, cost, provider interaction, side effect monitoring, travel, stigma, and work schedule), a decision-making process that was guided by a shared decision-making tool (flipbook). Following this process, the study nurse would review the stated preferences with the participant and ask for their final informed choice of delivery strategy, reminding them that they had the option to switch strategies at any point if desired.

## Data collection

Baseline demographic and clinical surveys were administered to all participants before their first 3HP dose. A completion survey was conducted during the final dose visit. 3HP dosing information was extracted from the 99DOTS server. Routine clinic staff, with no extra training, performed all other study procedures. No study-related attempts were made to contact participants for missed doses or adverse events. Both study and clinic staff logged weekly time requirements for their duties, and clinical activities were observed using time-and-motion surveys. Time data were transferred to human resource cost based on the national average level of each type of staff's hourly salary. Study staff completed monthly budgetary reviews of bills and receipts for resource-use data, including overhead costs, study drugs, drug import fees, office supplies, internet charges, and lab consumables. Cost of transport reimbursement was calculated based on the total number of patients and their number of visits in each arm.

## Outcomes

The primary outcome measured the reach (acceptance) and fidelity (treatment completion) of 3HP, defined as the proportion of participants who took at least 1 dose of 3HP (acceptance) and completed at least 11 of 12 doses of 3HP within 16 weeks of treatment initiation (completion) [7,9,10]. Prespecified subgroup analyses included age group, sex, time on ART, and prior TB or TB infection treatment experience. Prespecified secondary outcomes were 3HP acceptance (taking at least 1 dose) and 3HP completion among those who accepted (prespecified as the trial per-protocol analysis). Additional prespecified secondary outcomes reported here include the following: adverse events leading to 3HP discontinuation, the fidelity of intervention and delivery strategy implementation (reimbursement metrics, time spent at clinical visits/time spent on 3HP-related activities, time spent completing the shared decision-making tool, proportion of IVR check-in calls, dosage reminders, and appointment reminders sent, proportion of participants who received follow-up/support actions following response to IVR

check-ins or missed appointments/doses, and proportion of doses confirmed using the 99DOTS platform), the acceptability of delivery strategies (median scores for domains within the patient satisfaction questionnaire regarding services received throughout the course or 3HP treatment, median scores for domains within the shared decision-making questionnaire regarding the shared decision-making process), and the per-patient cost of 3HP delivery from the health system perspective. Additional secondary outcomes that will be reported separately include cost-effectiveness, patient costs, thematic results from patient and healthcare provider in-depth and key informant interviews, self-reported patient barriers to TB preventive services, provider- and clinic-level barriers to 3HP delivery, and data on active TB screening and TB incidence up to 2 years posttreatment [12]. Data on efficacy endpoints are still being collected.

## Statistical analysis

We estimated sample size based on a minimum clinically important difference of 10% in 3HP acceptance and completion between DOT and informed choice arms, which is consistent with previous implementation trials of 3HP delivery strategies [9]. To be conservative, we applied a Bonferroni correction for 2 independent comparisons (DOT versus Choice and SAT versus Choice). Assuming 10% loss between consent and allocation, we determined that 552 participants per arm ($N = 1,656$ total) would achieve 90% power (two-sided $\alpha = 0.025$). This sample size also provided 85% power to detect a point estimate of at least 80% 3HP acceptance and completion in the Choice arm, assuming a true level of 85%.

Following our prespecified statistical analysis plan (S2 Text), we calculated 3HP acceptance and completion in each arm as a proportion with exact binomial confidence interval (CI). To achieve the target of at least 80% acceptance and completion, we assessed whether the lower bound of the Bonferroni-corrected 97.5% CI exceeded 0.80 in any of the 3 arms. All participants were analyzed in the treatment arms to which they were randomized. We compared proportions between arms by calculating unadjusted prevalence ratios and using Fisher's exact test to determine statistical significance (two-sided $\alpha = 0.025$). As per our statistical analysis plan, we conducted an as-treated analysis, where participants in the Choice arm were allocated to the DOT or SAT arm based on their initial choice of strategy, and the DOT and SAT arms were compared using the same approach described above. We estimated the total per-patient cost of each delivery strategy from the health system perspective by compiling cost data from time and motion surveys, weekly time logs, and monthly budgetary reviews. Costs were estimated in 2021 Ugandan Shillings and converted to US dollars using the average exchange rate in 2021 (1 USD = 3,587.05 USh) [18].

All statistical analyses were performed using Stata version 16 (Stata Corporation, USA).

## Results

Of 2,813 PLHIV screened between July 13, 2020, and July 8, 2022, 1,656 were eligible and randomized. Follow-up continued until September 29, 2022. One participant was erroneously rerandomized (facilitated DOT arm) after an initial enrollment; data from their second randomization was excluded. Median participant age was 42 years (interquartile range [IQR]: 36 to 48); 1,122 (67.8%) were female; and median ART experience was 9.0 years (IQR: 5.6 to 12.5). Baseline characteristics were balanced across study arms (Table 1). In the Choice arm, 370 participants (67.0%) initially preferred facilitated DOT. Choice of delivery strategy did not differ across educational strata or age; however, males were more likely than females to choose DOT (S1 Table). During 3HP treatment, 11 Choice participants switched strategies (4 from SAT to DOT, 6 from DOT to SAT, 1 from DOT to SAT, then back to DOT).

Among the 1,655 PLHIV included in the primary analysis, 81 (4.9%) participants who initiated 3HP completed fewer than 11 doses within the allotted 16-week treatment period and were classified as not reaching the primary outcome. Posttreatment survey data among a sample ($n$ = 73) of these 81 participants suggested side effects, transport, and work-related challenges were the most important barriers (S2 Table). Additionally, 23 (1.4%) other participants did not accept or complete 3HP treatment, including 18 for whom a routine clinician discontinued 3HP due to adverse events ($N$ = 14), pregnancy ($N$ = 2), drug–drug interaction ($N$ = 1), or COVID-19 ($N$ = 1); 4 who never initiated 3HP; and 1 who died for reasons unrelated to the trial.

The proportion of participants who accepted and completed 3HP exceeded 0.80 for all 3 delivery strategies ($p$ < 0.001): 521 participants in both the facilitated DOT (0.95; 97.5% CI [0.92, 0.97]) and Choice arms (0.94; 97.5% CI [0.92, 0.96]), and 509 (0.92; 97.5% CI [0.89, 0.95]) in the facilitated SAT arm (Fig 2). We found no evidence that the proportion accepting and completing 3HP differed when comparing SAT to DOT (prevalence ratio [PR] 0.98; 97.5% CI [0.94, 1.01] $p$ = 0.118), Choice to DOT (PR 1.00; 97.5% CI [0.97, 1.03] $p$ = 0.901), or Choice to SAT (PR 1.02; 97.5% CI [0.99, 1.06] $p$ = 0.149) (Fig 3).

Of the 1,655 participants included in the primary analysis, 1,651 (99.8%) accepted 3HP. Four participants did not initiate 3HP (3 in the Choice arm and 1 in the DOT arm). There was no strong evidence that 3HP acceptance differed between arms: SAT versus DOT (PR = 1.00; 97.5% CI [0.99, 1.01] $p$ = 0.317), Choice versus DOT (PR 1.00; 97.5% CI [0.99, 1.00] $p$ = 0.317), or Choice versus SAT (PR 0.99; 97.5% CI [0.99, 1.00] $p$ = 0.083; Fig 3). 3HP completion among those who accepted treatment (per-protocol analysis) was high across arms with no evidence of between-arm differences (Fig 2). Results from the as-treated analysis allocating participants in the Choice arm to their selected strategy were similar (PR = 0.98; 97.5% CI [0.95, 1.01] $p$ = 0.077).

Fourteen participants discontinued 3HP due to adverse events: 7 in the SAT arm, 4 in the Choice arm (all 4 chose DOT), and 3 in the DOT arm (S3 Table). Treatment discontinuation due to an adverse event did not differ when comparing SAT versus DOT (risk difference [RD] 0.72%; 97.5% CI [−0.56%, 2.00%] $p$ = 0.206), Choice versus SAT (RD −0.54%; 97.5% CI [−1.88%, 0.81%] $p$ = 0.368), or Choice versus DOT (RD 0.18%; 97.5% CI [−0.89%, 1.26%] $p$ = 0.703) (S4 Table).

Subgroup analyses demonstrated no evidence of differences between any 2 treatment arms (S3 and S4 Figs). Exploratory subgroup analyses found some evidence suggesting that, in the DOT arm, participants with ≥9 years of ART experience had higher acceptance and completion compared to those with <9 years (RD 4.10%; 97.5% CI [−0.19%, 8.39%] $p$ = 0.032). The average per-patient cost to the health system was $139 for DOT, $89 for SAT, and $123 for informed choice. Drugs and transport reimbursement accounted for 76% to 85% of all health system costs (Fig 4).

Fidelity to 3HP treatment and delivery strategy components was high overall and across arms. Overall, 99.4% of expected transport reimbursements were delivered (S5 Table); median time spent at the clinic for 3HP refill or DOT visits was short (median: 10 minutes, IQR: 5 to 20) (S6 Table); >85% of expected IVR check-in calls and appointment reminders were sent to participants (S7 Table); and participants receiving 3HP by SAT confirmed >85% of all expected doses via the 99DOTS platform (S8 Table). Similarly, patient satisfaction with 3HP treatment and delivery strategies was high, with median satisfaction being at the highest level for all questions asked among participants in the Choice arm (S9 and S10 Tables).

**Table 1. Participant baseline characteristics, by study arm (N = 1,655)[a].**

| | Facilitated DOT (N = 551) | Facilitated SAT (N = 552) | Informed Choice (N = 552) |
|---|---|---|---|
| **Initial choice of delivery strategy** | | | |
| Facilitated DOT | - | - | 370 (67.0%) |
| Facilitated SAT | - | - | 182 (33.0%) |
| **Age** | 42 (36–48) | 42 (36–48) | 42 (36–48) |
| **Female sex** | 378 (68.6%) | 375 (67.9%) | 369 (66.9%) |
| **Education** | | | |
| None | 34 (6.2%) | 56 (10.1%) | 49 (8.9%) |
| Primary | 271 (49.2%) | 245 (44.4%) | 266 (48.2%) |
| Secondary | 209 (37.9%) | 201 (36.4%) | 190 (34.4%) |
| Tertiary/Vocational | 20 (3.6%) | 32 (5.8%) | 33 (6.0%) |
| University/Graduate School | 17 (3.1%) | 18 (3.3%) | 14 (2.5%) |
| **Employment status[b]** | | | |
| Unemployed | 65 (11.8%) | 63 (11.4%) | 63 (11.5%) |
| Hired worker | 143 (26.1%) | 135 (24.5%) | 119 (21.8%) |
| Self-employed worker | 293 (53.4%) | 311 (56.3%) | 312 (57.1%) |
| Other | 48 (8.7%) | 43 (7.8%) | 52 (9.5%) |
| **Household size** | 4 (2–5) | 4 (3–5) | 4 (2–5) |
| **Multidimensional Poverty Index[c]** | | | |
| Not vulnerable to multidimensional poverty | 231 (41.9%) | 252 (45.7%) | 236 (42.8%) |
| Vulnerable to multidimensional poverty | 199 (36.1%) | 190 (34.4%) | 194 (35.1%) |
| Multidimensionally poor | 94 (17.1%) | 93 (16.9%) | 98 (17.8%) |
| Severely multidimensionally poor | 27 (4.9%) | 17 (3.1%) | 24 (4.4%) |
| **Travel time (minutes) to clinic** | 65 (40–105) | 60 (40–90) | 60 (39.5–90) |
| **Prior treatment for tuberculosis (TB) or TB infection** | 104 (18.9%) | 108 (19.6%) | 89 (16.1%) |
| **Time (years) on ART** | 9.0 (5.8–12.4) | 9.1 (5.6–12.5) | 9.1 (5.5–12.7) |
| **ART regimen** | | | |
| Dolutegravir + Lamivudine + Tenofovir | 482 (87.5%) | 489 (88.6%) | 480 (87.0%) |
| Tenofovir + Lamivudine + Efavirenz | 40 (7.3%) | 29 (5.3%) | 43 (7.8%) |
| Other[d] | 29 (5.3%) | 34 (6.2%) | 29 (5.3%) |
| **CD4+ T cell count (cells/μL)[e]** | 495 (325–682) | 447.5 (294–633) | 471 (332–656) |
| **Viral load <1,000 copies/mL[f]** | 541 (98.2%) | 545 (98.7%) | 547 (99.1%) |
| **Body Mass Index (kg/m$^2$)** | 25.4 (21.6–28.9) | 25.3 (22.2–29.5) | 24.9 (22.0–29.3) |

ART, antiretroviral therapy; DOT, directly observed therapy; MPI, multidimensional poverty index; SAT, self-administered therapy; TB, tuberculosis.

[a]Data are n (%) or median and interquartile range (IQR).

[b]n = 8 missing (DOT: n = 2; SAT: n = 0; Choice: n = 6).

[c]The global MPI examines deprivations across 10 indicators in dimensions of health, education, and standards of living, with those deprived in one-third or more of the 10 indicators counted as being multidimensionally poor. MPI scores can range from 0 to 1 and are classified as not vulnerable to multidimensional poverty (MPI score: 0–0.19), vulnerable to multidimensional poverty (MPI score: 0.20–0.32), multidimensionally poor (MPI score: 0.33–0.49), and severely multidimensionally poor (MPI score: ≥0.50).

[d]Other regimens included Abacavir + Lamivudine + Dolutegravir (n = 82), Lamivudine + Zidovudine + Dolutegravir (n = 8), Lamivudine and Zidovudine 150 mg/300 mg tablet (n = 1), Abacavir + Lamivudine + Efavirenz (n = 1).

[e]n = 30 missing (DOT: n = 5; SAT: n = 14; Choice: n = 11).

[f]Not detected defined as <1,000 copies/mL according to Ugandan guidelines; n = 2 missing (DOT: n = 1; SAT: n = 1; Choice: n = 0).

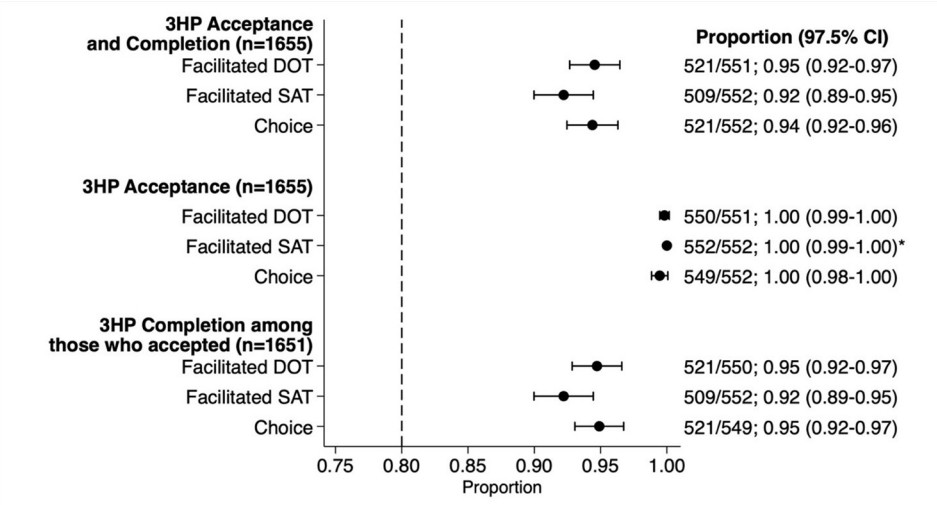

**Fig 2. 3HP acceptance and completion, by study arm.** The primary outcome of 3HP acceptance and completion was defined as the proportion of participants who took at least 11 of 12 doses of 3HP within 16 weeks of treatment initiation. Secondary outcomes included 3HP acceptance, defined as the proportion of participants who took at least 1 dose of 3HP, and 3HP completion, defined as the proportion of participants who took at least 11 of 12 doses of 3HP among those who accepted (prespecified as the trial per-protocol analysis). Point estimates of proportions are represented as solid circles with corresponding 97.5% CI error bars. The dotted vertical line at 0.80 represents the prespecified acceptance and completion threshold against which we assessed our coprimary hypothesis. (*) one-sided, 98.75% CI. CI, confidence interval; DOT, directly observed therapy; SAT, self-administered therapy; 3HP, 12 weeks of once-weekly isoniazid and rifapentine.

## Discussion

In this pragmatic type 3 effectiveness-implementation trial involving 1,655 PLHIV completed in Kampala, Uganda, we found high and comparable levels of 3HP acceptance and completion

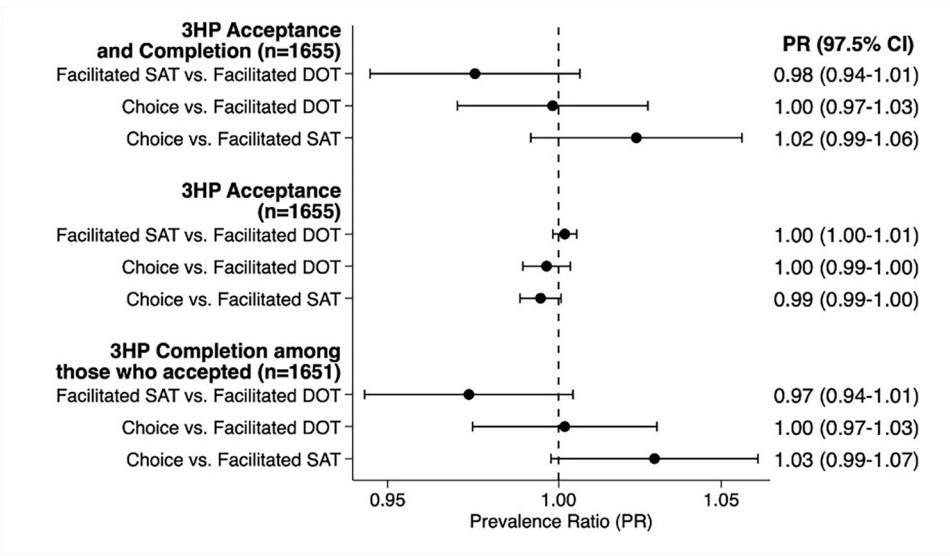

**Fig 3. Comparison of study outcomes across arms (*N* = 1,655).** Solid circles represent unadjusted PR point estimates, with 97.5% CIs depicted as error bars. Point estimates to the left of the vertical dotted line at a PR of 1.00 favor the first arm listed. CI, confidence interval; DOT, directly observed therapy; PR, prevalence ratio; SAT, self-administered therapy; 3HP, 12 weeks of once-weekly isoniazid and rifapentine.

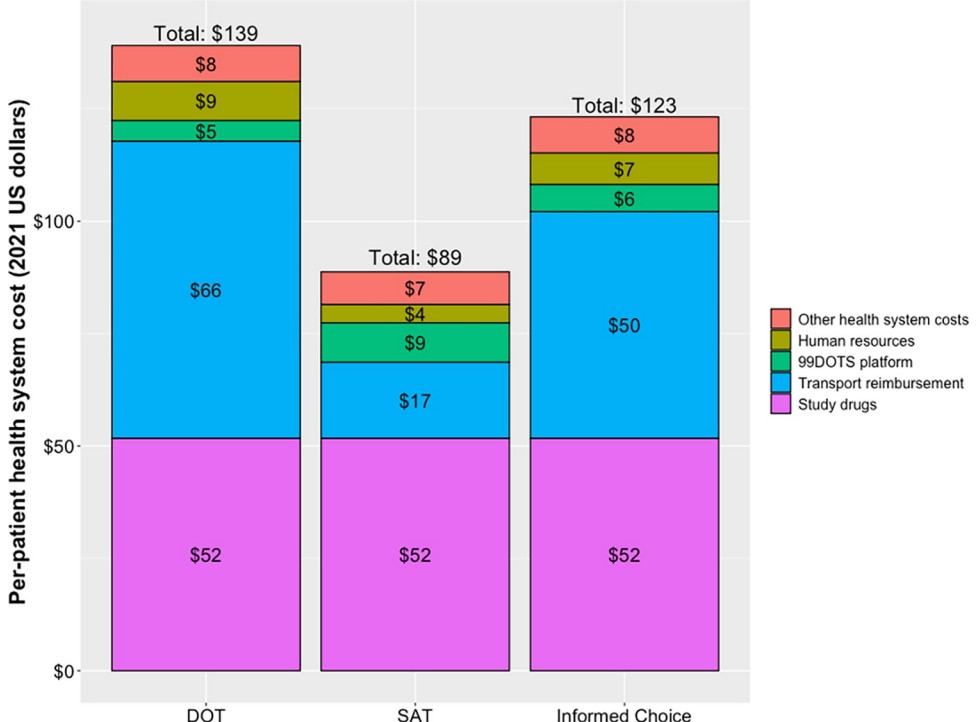

**Fig 4. Per-patient cost of facilitated 3HP delivery from the health system perspective.** Bars represent the average per-patient cost of facilitated 3HP from the health system perspective, according to individual components. "Other health system costs" include costs related to overheads, drug import fees, laboratory, office supplies, and internet charges (a, b). (a) 99DOTS is a technology whereby medications are packaged alongside toll-free phone numbers that are revealed when each dose is unpackaged, enabling patients to make toll-free calls to confirm medication dosing. Clinic staff can remotely access patient adherence data through a web dashboard. IVR reminders, check-in phone calls, and two-way messaging are also core features of the platform that enable real-time identification of patients who miss doses for further follow-up and monitoring of potential side effects. (b) Human resource costs included salaries of pharmacy technicians, lab technicians, clinicians, and entry-level and manager-level research staff. Hourly wages were calculated from government salary scales but adjusted to the project team's structure and allocated to activities based on direct observation (time and motion studies). DOT, directly observed therapy; IVR, interactive voice response; SAT, self-administered therapy; 3HP, once-weekly isoniazid and rifapentine taken for 3 months.

across all delivery strategies tested: facilitated DOT, facilitated SAT, and informed choice. Acceptance and completion rates reached 93.7% overall, surpassing our 80% target in all 3 arms. To the best of our knowledge, these levels are among the highest achieved in the context of routine HIV/TB care in sub-Saharan Africa. Our findings highlight that, with appropriate supports to overcome known barriers [12], very high acceptance and completion of short-course TPT can be attained within routine HIV/AIDS care, regardless of direct observation.

In our trial, completion rates were comparable to those achieved in the WHIP3TB trial conducted in South Africa (90.5%), Ethiopia (95.4%), and Mozambique (82.2%), where trained research staff administered 3HP [10]. In contrast, our completion rate was much higher than the 50% achieved with SAT in the South African iAdhere trial site [9]. However, it is possible in all 3 trials that actual adherence was lower with SAT as pill ingestion was not confirmed. A subgroup analysis of the WHIP3TB trial data, focusing on those who received 3HP facilitated by electronic medication monitoring, found lower treatment completion (83.5%) than the overall completion rate assessed by pill count alone [10]. In our trial, adherence was assessed by both self-report via the 99DOTS platform and pill counts at weeks 6 and 12. We also demonstrate that 3HP was well tolerated, with <1% of participants experiencing adverse events

resulting in treatment discontinuation. These findings add to the large evidence base demonstrating the safety/tolerability of 3HP for PLHIV taking efavirenz-based ART [7,19].

Our formative work identified 3 key barriers to 3HP completion: poor understanding of the need for preventive therapy, time and costs of accessing care, long wait times for medications, and difficulty incorporating a new medication into established routines. We addressed these barriers in all delivery strategies with counselling flipbooks designed to facilitate understanding of the importance of TPT for PLHIV, reimbursements covering transportation costs, streamlined clinic visits, and IVR phone call reminders [12]. Survey data reflect that the barriers related to cost and transportation remained a challenge for some participants who were unsuccessful in completing 3HP. However, the overall high level of 3HP acceptance and completion highlights the importance of identifying barriers through formative work and addressing those barriers during implementation.

Equivalent effectiveness and successful implementation of all strategies make cost a crucial consideration. Facilitated SAT is substantially cheaper to the health system than facilitated DOT ($89 versus $139 per patient course), primarily due to fewer clinic visits. The cost difference between DOT and SAT would likely be even higher if people living >25 kilometers of the study clinic were included in the trial, and incorporating participant-level costs (e.g., lost wages, childcare requirements for clinic visits) is likely to only further support facilitated SAT as a cheaper option than facilitated DOT. Shared decision-making in the Choice arm added only marginally to the total cost of 3HP delivery.

Our study's strengths include strong reliance on implementation science principles and theory-informed frameworks [11,17,20], active involvement of PLHIV and key stakeholders in trial design, and integration of 3HP delivery into routine care, increasing relevance and potential for future scale-up. Person-centeredness was a core aspect of the delivery strategy design, aligning with calls to enhance outcomes while also respecting human dignity [1,21]. Informed choice was introduced as a person-centered approach, showing feasibility and high completion levels when considering people's lifestyle and preferences.

Despite its strengths, this study also had some limitations. Although we conducted and reported an interim analysis of the primary outcome aggregated across arms [22], the minimal difference in the aggregate proportion accepting and completing 3HP in the interim versus final analysis (0.93 versus 0.94, respectively) and the lack of difference between trial arms suggest that the interim analysis did not have a major impact on final trial results. This was a single center trial that excluded people living further than 25 kilometers from the clinic and those who did not own/have access to a phone. As with any implementation trial, the tested strategies are likely not to be feasible to implement for all individuals in all settings. For example, individuals without cell phones cannot participate in cell phone–based reminder systems; similarly, people who live very far from a clinic are unlikely to be able to participate in DOT. As such, our findings of high acceptance and completion should be interpreted as reflective of what might be achievable among people who could participate in these interventions, not among all PLHIV eligible for TPT in any given clinic. Similarly, some of the delivery strategy components we evaluated (such as transport reimbursements and IVR phone calls) may not be considered feasible in other clinics. However, broader consideration of such costs (which are often lower than the costs of drugs) in health budgets is needed for effective implementation of novel interventions. Lastly, while the barriers to 3HP completion targeted here are likely to be relevant in many settings, further contextual adaptation may be required for facilitated 3HP implementation.

In conclusion, this trial demonstrated high 3HP acceptance and completion among PLHIV in a programmatic setting using theory-informed delivery strategies designed to address known barriers. Policymakers in similar high-burden settings can utilize these findings to

inform implementation strategies for short-course TPT (including 3HP) likely to maximize uptake and treatment completion.

## Supporting information

**S1 Text. Trial inclusion and exclusion criteria.**
(DOCX)

**S2 Text. Trial statistical analysis plan.**
(PDF)

**S1 Fig. Example of participant 99DOTS-based adherence calendars, including a participant on directly observed therapy (DOT) and a participant on self-administered therapy (SAT).**
(DOCX)

**S2 Fig. Self-administered therapy participant pill packs.**
(DOCX)

**S3 Fig. Subgroup analyses of the primary trial endpoint (acceptance and completion of 3HP).**
(DOCX)

**S4 Fig. Subgroup analyses comparing 3HP acceptance and completion across trial arms.**
(DOCX)

**S1 Table. Comparison of choice of delivery strategies across key participant characteristics.**
(DOCX)

**S2 Table. Reasons for stopping 3HP treatment.**
(DOCX)

**S3 Table. Adverse events leading to 3HP discontinuation.**
(DOCX)

**S4 Table. Pairwise comparisons of treatment discontinuation due to an adverse event as unadjusted odds ratios and unadjusted risk differences with corresponding 97.5% confidence intervals (CIs).**
(DOCX)

**S5 Table. Reimbursement metrics.**
(DOCX)

**S6 Table. Time spent on 3HP-related activities.**
(DOCX)

**S7 Table. 99DOTS implementation metrics.**
(DOCX)

**S8 Table. Confirmation of self-administered doses via 99DOTS.**
(DOCX)

**S9 Table. Patient Satisfaction Survey Results.**
(DOCX)

**S10 Table. Responses to Shared Decision-Making Questionnaire.**
(DOCX)

**S1 CONSORT Checklist. CONSORT 2010 checklist of information to include when reporting a randomized trial.**
(DOC)

**S1 Data. Supporting Information Trial Data.**
(XLSX)

## Acknowledgments

We are grateful to the staff and clients at the Mulago Immune Suppression Syndrome (ISS) HIV/AIDS clinic for their time and participation in the study.

## Author Contributions

**Conceptualization:** Fred C. Semitala, Payam Nahid, Robert Belknap, Moses R. Kamya, Margaret A. Handley, Patrick PJ Phillips, Anne Katahoire, Noah Kiwanuka, Achilles Katamba, David W. Dowdy, Adithya Cattamanchi.

**Data curation:** Jillian L. Kadota, Allan Musinguzi, Fred Welishe, Anne Nakitende, Lydia Akello, Lynn Kunihira Tinka, Jane Nakimuli, Joan Ritar Kasidi, Opira Bishop.

**Formal analysis:** Fred C. Semitala, Jillian L. Kadota, Suzan Nakasendwa, Yeonsoo Baik, Patrick PJ Phillips, Noah Kiwanuka, David W. Dowdy, Adithya Cattamanchi.

**Funding acquisition:** Fred C. Semitala, David W. Dowdy, Adithya Cattamanchi.

**Investigation:** Fred C. Semitala, Amanda Sammann, Payam Nahid, Robert Belknap, Moses R. Kamya, Margaret A. Handley, Patrick PJ Phillips, Anne Katahoire, Christopher A. Berger, Noah Kiwanuka, Achilles Katamba, David W. Dowdy, Adithya Cattamanchi.

**Methodology:** Fred C. Semitala, Jillian L. Kadota, Allan Musinguzi, Yeonsoo Baik, Devika Patel, Amanda Sammann, Payam Nahid, Robert Belknap, Moses R. Kamya, Margaret A. Handley, Patrick PJ Phillips, Anne Katahoire, Christopher A. Berger, Noah Kiwanuka, Achilles Katamba, David W. Dowdy, Adithya Cattamanchi.

**Project administration:** Fred C. Semitala, Jillian L. Kadota, Allan Musinguzi, Anne Katahoire, Christopher A. Berger, Achilles Katamba, David W. Dowdy, Adithya Cattamanchi.

**Resources:** Fred C. Semitala, Achilles Katamba.

**Supervision:** Fred C. Semitala, Allan Musinguzi, Anne Katahoire, Achilles Katamba, David W. Dowdy, Adithya Cattamanchi.

**Visualization:** Jillian L. Kadota, Suzan Nakasendwa, Yeonsoo Baik, Patrick PJ Phillips, Noah Kiwanuka.

**Writing – original draft:** Fred C. Semitala, Jillian L. Kadota, David W. Dowdy, Adithya Cattamanchi.

**Writing – review & editing:** Fred C. Semitala, Jillian L. Kadota, Allan Musinguzi, Fred Welishe, Anne Nakitende, Lydia Akello, Lynn Kunihira Tinka, Jane Nakimuli, Joan Ritar Kasidi, Opira Bishop, Suzan Nakasendwa, Yeonsoo Baik, Devika Patel, Amanda Sammann, Payam Nahid, Robert Belknap, Moses R. Kamya, Margaret A. Handley, Patrick PJ Phillips, Anne Katahoire, Christopher A. Berger, Noah Kiwanuka, Achilles Katamba, David W. Dowdy, Adithya Cattamanchi.

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
