## [Editor Report · Decision Letter 0]

21 Aug 2023

Dear Dr Cattamanchi, 

Thank you for submitting your manuscript entitled "Comparison of three optimized delivery strategies for completion of Isoniazid-Rifapentine (3HP) for tuberculosis prevention among people living with HIV: a randomized trial" for consideration by PLOS Medicine.

Your manuscript has now been evaluated by the PLOS Medicine editorial staff as well as by an academic editor with relevant expertise and I am writing to let you know that we would like to send your submission out for external peer review.

Please re-submit your manuscript within two working days, i.e. by Aug 23 2023 11:59PM.

Kind regards,

Katrien Janin, PhD

Senior Editor

PLOS Medicine

---

## [Decision Letter · Decision Letter 1]

17 Nov 2023

Dear Dr. Cattamanchi,

Thank you very much for submitting your manuscript "Comparison of three optimized delivery strategies for completion of Isoniazid-Rifapentine (3HP) for tuberculosis prevention among people living with HIV: a randomized trial" (PMEDICINE-D-23-02381R1) for consideration at PLOS Medicine. 

Your paper was discussed among all the editors here. It was also discussed with an academic editor with relevant expertise, and sent to independent reviewers, including a statistical reviewer. The reviews are appended at the bottom of this email and any accompanying reviewer attachments can be seen via the link below:

[LINK]

We will not be able to accept the manuscript for publication in the journal in its current form, but we would like to consider a revised version that addresses the reviewers' and editors' comments. We cannot make any decision about publication until we have seen the revised manuscript and your response, and we plan to seek re-review by one or more of the reviewers. 

We expect to receive your revised manuscript by Dec 08 2023 11:59PM. Please email us (plosmedicine@plos.org) if you have any questions or concerns.

We look forward to receiving your revised manuscript. 

Sincerely,

Katrien Janin, PhD

PLOS Medicine

plosmedicine.org

Comments from the academic editor:

It is an excellent paper but in order for this to not be just more data compared to before, further subanalysis would be valuable.

Comments from the editorial team: 

We are much in agreement with the academic editor and reviewer #2 that the secondary outcomes should be reported here; particularly given the noted study limitations and that the full results are similar to the interim analysis. We offer major revisions contingent on presentation of secondary outcomes.

—------------------------------------------------------------------------------------

GENERAL: 

Please provide 95% CIs and p values for all results where appropriate (including the abstract), check and amend throughout. We suggest reporting statistical information in the following format: ‘x’; (95% CI [‘y’,’ z’] p value) and use commas as opposed to hyphens (as these can be confused with negative values) to separate upper and lower bounds. 

For p values, please report as p<0.001 and where higher as 'p=0.002'. Please add the statistical method used to your method section. We also invite you to report p values to consistently to the third decimal digit - thousandths.

For in-text reference, citations are placed within square parentheses and should precede punctuation. Please amend throughout. (e.g line 107-108 should be: ‘Tuberculosis (TB) is a curable and preventable disease but remains a leading cause of death worldwide, especially among people living with HIV (PLHIV) [1].) Please amend throughout. 

STUDY DESIGN: 

Please complete the CONSORT checklist and ensure that all components of CONSORT are present in the manuscript. When completing the checklist, please use section and paragraph numbers, rather than page numbers.

TITLE: Please add locality (e.g. Uganda, or Kampala, Uganda) in the title. Please also specify in the title that this study is a single centre study.

ABSTRACT:

Please report your abstract according to CONSORT for abstracts, https://www.equator-network.org/reporting-guidelines/consort-abstracts/ and following the PLOS Medicine abstract structure (Background, Methods and Findings, Conclusions). Please remove all other subheaders.

Abstract Methods and Findings:

Please include in the study design that the study is single centre.

Please quantify the main results (with 95% CIs and p values).

In the last sentence of the Abstract Methods and Findings section, please describe the main limitation(s) of the study's methodology.

AUTHORS SUMMARY:

Ideally each sub-heading should contain 2-3 single sentence, concise bullet points containing the most salient points from your study.

In the final bullet point of ‘What Do These Findings Mean?’ Please include the main limitations of the study in non-technical language.

ACKNOWLEDGMENTS/ DECLARATIONS

Please remove all statements apart from acknowledgements, author contributions and abbreviations from the end of the main manuscript and include these only in the relevant parts of the manuscript submission form. Funding, competing interest, and data availability will be compiled as metadata.

SUPPORTING INFORMATION has not been checked by the editor.

Comments from the reviewers:

Reviewer #1: "Comparison of three optimized delivery strategies for completion of Isoniazid-Rifapentine (3HP) for tuberculosis prevention among people living with HIV: a randomized trial" reports the outcome of a pragmatic three-arm parallel-group randomized trial with about 550 participants in each arm, on acceptance (and adherence) to short-course tuberculosis (TB) prevention treatment, through either of directly observed therapy (DOT), facilitated self-administered therapy (SAT) or an informed choice between DOT and SAT. The main finding was that all three regimens had high acceptance and completion rates (>0.92).

While generally clearly written and extensively documented with implications for TB prevention administration, some issues might be considered:

1. In Line 136, a number of exclusion criteria are stated. In particular, the exclusion for people <40kg might be explained further. What happened to would-be participants weighing less than 40kg?

2. In Line 213, the fidelity rate is defined as the completion of at least 11 of 12 doses of 3HP within 16 weeks of treatment initiation. The definition might be briefly justified.

3. In Line 223, a minimum clinically important difference of 10% is stated. It might be briefly clarified if this was established by custom, empirical justification or prior literature etc.

4. In Line 231, the Supplement number might be stated.

5. In Line 271, it is stated that "...73 participants who initiated 3HP missed six or more doses and were classified as not reaching the primary outcome (unable to complete 11 doses within 16 weeks)". The discrepancy between the actual applied condition (missed six or more doses) and the original definition (complete 11 [of 12 doses, from Line 213]) might be explained.

6. Participant-reported reasons for missing multiple doses might be briefly discussed in the Discussion section, if possible.

7. In Line 347, it is stated that the adherence rate for WHIP3TB might be suggested to be lower than estimated by completion rate, once pill count was considered. Then, it is stated that "...we demonstrate high levels of 3HP completion by SAT facilitated by the 99DOTS adherence technology". It might be clarified as to whether the 99DOTS technology addresses the issue of adherence rate not complementing the completion rate, i.e. how doe 99DOTS solve the problem of pills/doses reported as being taken, but not actually taken?

8. The calculation/data sources (e.g. from surveys?) for the costings reported in Figure 4 might be briefly described.

Reviewer #2: Review for PLoS Medicine

Manuscript Number: PMEDICINE-D-23-02381R1

Full Title: Comparison of three optimized delivery strategies for completion of Isoniazid-Rifapentine (3HP) for tuberculosis prevention among people living with HIV: a randomized trial

October 20, 2023

The authors conducted a randomized controlled implementation science trial to determine the optimal delivery strategies for TB preventive therapy (3HP) in Uganda. The team had two distinct intervention arms, and a third trial arm that allowed people to make an informed choice. Overall, the acceptance and completion rates were high, and there were no differences among the three groups.

The authors have already presented and published an analysis of acceptance and completion rates for this clinical trial cohort (PLoS Med. 2021/12/17 ed. 2021; Dec;18(12):e1003875). The authors should comment on whether and how the presentation/publication of the interim results could have influenced the remainder of the trial.

In addition, the 3HP adoption, implementation, and outcomes measures of TB incidence at one-year are not being reported in this paper. Since the interim analyses showed high levels of acceptability and completion, these secondary outcomes would be helpful to be reported in this paper to help inform programmatic decisions for 3HP delivery programs. Currently, the additional analyses, stratified by study arm, provide limited additional value. 

A major limitation of the study is being a single-site clinical study at one urban HIV clinic in Kampala. Given this is an IS study, using results from a single trial site to extrapolate programmatic approaches to the rest of Uganda and/or beyond can be problematic. The rather high treatment completion rates observed in this study also raises concerns that the population (and outcomes) may not be representative of other populations receiving 3HP. The authors could expand on how these results may be relevant (or not) to other settings.

The study did not include a control group that was not receiving adherence support. Therefore, differences above the baseline acceptance and completions rates were not assessed. Thus, the study does not address whether adherence support is still needed with the 3HP regimen. This should be addressed in the Discussion section.

Another limitation was the reliance on pill counting among the SAT group. Since obtaining a pill from a pack may not equate to pill ingestion, this outcome was not as rigorous as other possible measurements. However, given the lack of outcome difference, this issue was unlikely to have been problematic in this cohort.

The study appeared to use a composite outcome of reach and fidelity. The definition for the acceptance component was defined in Figure 2, but should be moved to the Methods section. Furthermore, reporting an outcome of acceptance is not particularly helpful for an RCT with informed consent (as evidenced in Figure 2). This outcome could be removed from Figure 2, Fig 3, and elsewhere.

Similarly, The 'informed choice' column can be removed from Figure 4, since this is a weighted average of participants who joined the 2 interventions. While this may be a representative cost for this study arm, it does not appear to represent raw data collected within the choice intervention.

Overall, there was a high rate of exclusion for people not residing in the local area. This likely had an impact on the high completion rates, and also limits generalizability of study findings. The impact of this exclusion criteria and the impact of interpretation should be discussed.

Reviewer #3: I commend the authors on conducting a well-designed, thoughtful and clinically relevant prospective 3-arm type 3 hybrid effectiveness-implementation study into three delivery strategies (DOT, self administered therapy or informed choice) for 3HP treatment of latent TB integrated into the workflow of a single large outpatient clinic serving persons with HIV in Kampala, Uganda. I agree with the authors that making shorter effective TB preventative therapy (latent TB) regimens such as 3HP more widely available among persons with HIV in high burden settings is a key component of the END TB strategy, and as noted by the authors in the introduction and discussion sections previous studies including iAdhere and WHIP3TB have found variable rates of treatment completion with regard to self-administered 3HP in other clinical settings in Africa. Among the 1655 participants in the primary analysis, those in all 3 arms achieved very high rates of acceptance and completion of 3HP treatment. The inclusion criteria and exclusion criteria are clearly laid out. Persons with HIV who started ART within the preceding 3 months were excluded and the authors should specify why this cutoff was used- I am slightly unclear as to whether the concern was for drug interactions between rifapentine and ART or for IRIS/unmasking of opportunistic infections as potential confounders of study outcomes. The exclusion of otherwise eligible participants who either did not have a cellphone to receive SMS reminders/automated calls and the exclusion of participants living >25km from the clinic do limit the generalizability of the findings and procedures used with regard to other settings in sub-Saharan Africa, where patients may need to travel long distances to receive care and where availability of cellphone coverage can vary, especially in rural settings. While the authors hint at this in the comments about generalizability to rural settings in the discussion section, I think these are important considerations that would strengthen the discussion. Additionally, I note from the protocol that only those participants who could consent in Luganda or English were eligible, and that another limitation in terms of implementing the phone-based reminder strategies used in this research study more broadly in other resource limited settings may be the need for materials in multiple languages and strategies to mitigate barriers based on literacy. This is where I feel that the single center approach used is something of a limitation, as implied by the authors in the penultimate paragraph of the discussion. The authors do acknowledge that reimbursement of travel and some of the other strategies to mitigate barriers to treatment completion may not be feasible in other settings and that the the barriers may differ in different settings. Overall, while this study failed to show that informed choice performed significantly better than DOT or self-administered therapy for 3HP in this setting, the authors clearly demonstrate in their results that self-administered therapy with 3HP is associated with high rates of acceptance and completion in a routine HIV clinic setting in sub-Saharan Africa. I look forward to the promised additional analyses of secondary endpoints, such as TB incidence at 1 year and cost-effectiveness analysis as mentioned in the 'Outcomes' paragraph of the methods section. 

Reviewer #4: I appreciate the opportunity to review this manuscript. The assessment of delivery strategies to improve the acceptance and completion of preventive therapy in people living with HIV is an incredibly important topic being studied in a high-risk population. I praise the use of human centered design methods and person-centered care principles for the optimization of delivery of 3HP. The high acceptance and completion of 3HP is incredibly promising, especially because of the authors' work done locally to integrate these delivery strategies into routine care and the potential for scale-up. I applaud the authors for an incredibly well-written manuscript that is clear to follow and thorough in its description. All tables and figures are clear and concise. While I fully support this manuscript for publication, I have a few minor suggestions for the final version: 

1) Did the authors assess completion results separately for individuals who received loose pills early on in the trial vs fixed-dose combination pills later in the trial? Do they anticipate any difference in acceptance or completion based on this?

2) Did the authors consider also conducting analysis for all those who were assigned a strategy (DOT or SAT) compared to all those who were provided a choice (despite which strategy they chose) to assess whether just having a choice in their healthcare management improved their acceptance or completion? 

3) The article might benefit from a few more details in the methods section about how costs were collected and calculated. 

4) Some information in the discussion re: formative work identifying barriers that were incorporated into the delivery strategies would be beneficial to describe in the context of the description of the strategies in the methods section. 

Reviewer #5: I considered the work of Prof. Adithya Cattamanchi and colleagues to be an extension of their published interim analysis, and it is a well-documented study. I would like the following explanations. 

1. Lines 136-142: The paragraph's exclusion criteria seem repetitive because Supplement 1 already mentions them.

2. Was the relationship between education level and the informed Choice 3HP preventative treatment examined by the author?

3. All three treatment arms have similar median and range ages. How could that be?

4. Why were there more female participants in all three arms of the study, despite the fact that those who were pregnant, breastfeeding a baby, or planning to get pregnant within the next 120 days were not eligible?

5. What is prior Tuberculosis? Patients with active tuberculosis, suspicion of active tuberculosis, and those who previously completed treatment for tuberculosis were not eligible for the study.

[LINK]

---

## [Decision Letter · Decision Letter 2]

19 Jan 2024

Dear Dr. Cattamanchi,

Thank you very much for re-submitting your manuscript "Comparison of three optimized delivery strategies for completion of Isoniazid-Rifapentine (3HP) for tuberculosis prevention among people living with HIV in Uganda: a single-center randomized trial" (PMEDICINE-D-23-02381R2) for review by PLOS Medicine.

I appreciate your detailed response to the editors' and reviewers' comments. I have discussed the paper with my colleagues and the academic editor, and it has also been seen again by four of the original reviewers. The changes made to the paper were satisfactory to the reviewers. As such, we intend to accept the paper for publication, pending your attention to the editorial comments below in a further revision. When submitting your revised paper, please once again include a detailed point-by-point response to the editorial comments.

[LINK]

In revising the manuscript for further consideration here, please ensure you address the specific points made by each reviewer and the editors. In your rebuttal letter you should indicate your response to the reviewers' and editors' comments and the changes you have made in the manuscript. Please submit a clean version of the paper as the main article file. A version with changes marked must also be uploaded as a marked up manuscript file. Please also check the guidelines for revised papers at http://journals.plos.org/plosmedicine/s/revising-your-manuscript for any that apply to your paper.

We ask that you submit your revision within 1 week (Jan 26 2024). However, if this deadline is not feasible, please contact me by email, and we can discuss a suitable alternative.

Please do not hesitate to contact me directly with any questions (aschaefer@plos.org). If you reply directly to this message, please be sure to 'Reply All' so your message comes directly to my inbox.

We look forward to receiving the revised manuscript.

Sincerely,

Alexandra Schaefer, PhD

On behalf of:

Katrien Janin, PhD

Associate Editor 

PLOS Medicine

Requests from Editors:

ABSTRACT

1) Abstract Background: Please provide the study question/hypothesis as the final sentence of the background section.

2) Please remove the trial registration details from the Methods and Findings section and add it at the end of the abstract as a separate information.

3) In the last sentence of the Abstract Methods and Findings section, please clearly describe the main limitation(s) of the study's methodology. Editorial suggestion: The main limitation of the study is that it was conducted in a single center, and multicenter studies are now needed to confirm the feasibility and and generalizability of facilitated 3HP delivery strategies in other settings.

AUTHOR SUMMARY

1) Under ‘Why Was This Study Done?’, we feel the second bullet point describes differences to two previous implementation trials without clearly stating the reason for conducting the current study. Please revise. Editorial suggestion: Two previous implementation trials have evaluated the delivery of 12 weeks of isoniazid and rifapentine (3HP) through self-administered therapy (SAT) in high-burden settings and found variable treatment completion rates, in both studies, 3HP dosing and treatment supervision were provided by research staff rather than routine health workers.

2) Please revise the bullet points under ‘What Did the Researchers Do and Find?’. We feel that a bullet point describing treatment options is currently missing and that the interpretation of these findings should be limited to question 3 (What Do These Findings Mean?). Editorial suggestion:

• In a pragmatic trial, 1,655 people living with HIV (PLHIV) at a high-volume clinic in Kampala, Uganda, received 3HP either by facilitated directly observed therapy (DOT) involving a pharmacy technician, facilitated self-administered therapy (SAT), or informed choice between facilitated DOT and facilitated SAT using a shared decision-making aid.

• We found high levels of 3HP completion in the context of routine HIV/AIDS care.

• The high completion rates were independent of therapy delivery strategy.

INTRODUCTION

1) ll.134-135: Please provide reference for the sentence starting “Scaling up TPT is crucial...”.

2) l.143: Please exchange ‘Both’ with ‘Two’.

METHODS AND RESULTS

1) ll.209-226: We feel that the detailed descriptions of facilitated directly observed therapy (DOT), facilitated self-administered therapy, and informed choice delivery strategies provided in Supplement 2 should be included in the main manuscript. Please revise accordingly and remove Supplement 2.

2) ll.240-250: In the Methods section under “Outcomes”, we ask that you include a complete list of secondary outcomes (it is not necessary to include all sub-bullet points as listed in the published protocol), followed by a statement about which of them are reported in this paper. Please also mention that data on efficacy endpoints are still being collected.

3) l.282: The terms gender and sex are not interchangeable (as discussed in https://www.who.int/health-topics/gender); please use the appropriate term. We suggest using the word 'female' here.

4) l.286: Given that Supplement 4 contains 8 tables, we suggest specifying the reference (here: Table 1 in Supplement 4). Please revise throughout the main manuscript.

5) l.313: “one who died in a motor vehicle accident.” - We suggest that this statement be reworded to read "one who died for reasons unrelated to the trial". Since the study was conducted in a single center and at a specific time, we are concerned that this may expose the deceased individual.

6) Per CONSORT, please present adverse events in a table and discuss whether or not adverse events are thought to be related to treatment.

DISCUSSION

1) l.424: “Informed choice was introduced as a novel, person-centered…” – please remove the word ‘novel’.

2) l.446: Please remove the word “extremely”.

FIGURES

1) Figure 1: Please define '3HP'. Please note that it is not necessary to define abbreviations within the figure and additionally list these abbreviations below the figure, either one is sufficient.

2) Figure 1: In the boxes directly above the numbers analyzed: In the SAT group, it is not necessary to indicate that "0 never initiated therapy", please remove.

3) Figure 4: We suggest adding footnotes detailing what the 99DOTS platform is and what human resources includes.

REFERENCES

When specifying the date of access, please write “accessed” instead of “cited”.

SUPPLEMENTARY MATERIAL

1) Please be sure to define 3HP where applicable.

2) Please be sure to define what the strategy “Choice” encompasses where applicable.

3) Supplement Figure 1: Please also define the meaning of the yellow boxes.

4) Supplement Figure 3: Please change ‘women’ and ‘men’ to ‘female’ and ‘male’.

5) Supplement Figure 4: Please change ‘women’ and ‘men’ to ‘female’ and ‘male’. Please mention in the figure description that the colours indicate the pre-specified subgroups (red = sex, blue = age, purple = time on ART, green = prior TB status).

SOCIAL MEDIA

To help us extend the reach of your research, please provide any X (formerly known as Twitter) handle(s) that would be appropriate to tag, including your own, your coauthors’, your institution, funder, or lab. Please respond to this email with any handles you wish to be included when we tweet this paper.

Comments from Reviewers:

Reviewer #1: We thank the authors for addressing our previous concerns.

Reviewer #2: The authors have been mostly responsive to the prior comments, and the manuscript is suitable for publication. However, given the dis-aggregated adherence measures were similar across 3 study arms, and the aggregated results have been previously published, this manuscript adds little value to the prior publication. The addition of the secondary outcomes was helpful, but also did not add much value to this manuscript.

Reviewer #3: The revisions and author responses have addressed all of my previous comments and concerns about this manuscript. The revised version is clearly written and improved by the incorporation of suggestions made by my fellow reviewers. In the Methods and Findings section of the abstract, consider the following very minor change for improved readability 'facilitated SAT using a shared decision-making aid' would perhaps be clearer as 'facilitated SAT using a shared decision-making aid (Choice)' to be consistent with the designation used later in the manuscript for the third group of participants.

Reviewer #4: I appreciate the thoughtful and comprehensive response to the reviewers' comments. All pending queries have been resolved and I support this manuscript for publication.

[LINK]

General Editorial Requests

---

## [Editor Report · Decision Letter 3]

2 Feb 2024

Dear Dr Cattamanchi, 

On behalf of my colleagues and the Academic Editor, I am pleased to inform you that we have agreed to publish your manuscript "Comparison of three optimized delivery strategies for completion of Isoniazid-Rifapentine (3HP) for tuberculosis prevention among people living with HIV in Uganda: a single-center randomized trial" (PMEDICINE-D-23-02381R3) in PLOS Medicine.

PRESS

Sincerely, 

Katrien G. Janin, PhD 

Senior Editor 

PLOS Medicine